# Structural characteristics of alpha-fetoprotein, including N-glycosylation, metal ion and fatty acid binding sites
Kun Liu[1,5], Cang Wu [2,5], Mingyue Zhu[1,5], Junnv Xu[1,3,5], Bo Lin [1], Haifeng Lin[3], Zhongmin Liu [2] ✉ & Mengsen Li [1,2,4] ✉

Alpha-fetoprotein (AFP), a serum glycoprotein, is expressed during embryonic development and the pathogenesis of liver cancer. It serves as a clinical tumor marker, function as a carcinogen, immune suppressor, and transport vehicle; but the detailed AFP structural information has not yet been reported. In this study, we used single-particle cryo-electron microscopy(cryo-EM) to analyze the structure of the recombinant AFP obtained a 3.31 Å cryo-EM structure and built an atomic model of AFP. We observed and identified certain structural features of AFP, including N-glycosylation at Asn251, four natural fatty acids bound to distinct domains, and the coordination of metal ions by residues His22, His264, His268, and Asp280. Furthermore, we compared the structural similarities and differences between AFP and human serum albumin. The elucidation of AFP's structural characteristics not only contributes to a deeper understanding of its functional mechanisms, but also provides a structural basis for developing AFP-based drug vehicles.

Alpha-fetoprotein (AFP) is a 69 kDa fetal serum glycoprotein primarily synthesized in the fetal liver and yolk sac, with a half-life of 4–5 days[1]. In addition to its high levels during pregnancy, AFP can also be detected at elevated concentrations in the plasma of patients with liver cancer, and AFP has been recognized as one of the earliest identified tumor markers[2]. Increasing evidences suggest that AFP plays an important biological role in the development of hepatocellular carcinoma (HCC). AFP promotes proliferation, invasion, and metastasis of HCC cells, inhibits apoptosis, and enhances the expression of stemness genes. These malignant behaviors are achieved by activating or inhibiting downstream target gene expression through signaling pathways such as cAMP-PKA, RA-RAR, Caspase3, PTEN, and PI3K/AKT/mTOR[3,4]. In addition, AFP has immunosuppressive functions. For example, AFP inhibits the metabolism of DC mitochondria, induces DC apoptosis[5], enhances the cytotoxicity of NK cells[6,7] and alters the ratio of CD4[+]/CD8[+] T cells, which inhibits T cell-mediated cytotoxicity[8]. Furthermore, AFP influences macrophage differentiation and phagocytic activity[9]. By participating in immune regulation, AFP promotes the immune escape of HCC cells[10], making it applicable in immunotherapy for HCC, such as AFP-based cancer vaccines[11] and AFP as a target for chimeric antigen receptor T-cell or T cell receptor T-cell therapy[12,13]. AFP plays an

essential role in HCC development through different mechanisms. However, the structural features of AFP have yet to be reported.

Since the discovery of AFP in 1956[14], extensive research has been conducted to investigate its structural and biochemical properties including protein stability, secondary structure, ligand binding, and two-dimensional imaging. Notably, glycoprotein variants of AFP were initially identified and isolated using neuraminidase and concanavalin A[15,16]. Subsequent investigations used various lectins to determine the types of sugar chains present in AFP and found distinct glycan compositions of AFP during pregnancy and tumorigenesis[17,18]. AFP-L3, a glycoform of AFP that specifically binds to lens culinaris agglutinin, has emerged as a novel tumor marker for HCC[19]. In 1990, the AFP glycopeptide structure was successfully identified using 1H-NMR[20]. Advancements in mass spectrometry and related techniques have subsequently facilitated the efficient identification of AFP glycopeptide types, contributing to the development of AFP as a tumor marker[21]. However, the effect of glycosylation on AFP function remains unclear. Studies have revealed that glycosylation enhances the production of recombinant AFP, but its impact on tumor cell growth and apoptosis is minimal[22]. In addition, AFP exhibits binding and transportation capabilities for biological molecules that are crucial for nutrient transport during

[1]Hainan Provincial Key Laboratory of Carcinogenesis and Intervention, Hainan Medical College, Haikou 571199 Hainan, PR China. [2]Department of Biology, School of Life Sciences, Southern University of Science and Technology, Shenzhen 518055 Guangdong, PR China. [3]Department of Medical Oncology, Second Affiliated Hospital, Hainan Medical College, Haikou 570023 Hainan, PR China. [4]Institution of Tumor, Hainan Medical College, Haikou 570102 Hainan, PR China. [5]These authors contributed equally: Kun Liu, Cang Wu, Mingyue Zhu, Junnv Xu. ✉e-mail: liuzm@sustech.edu.cn; mengsenli@163.com

embryonic development. Notably, various fatty acids (FA) have been isolated from umbilical cord blood-derived AFP[23]. In addition to its capacity to bind FA, AFP can form complexes with bilirubin. While the affinity of AFP towards bilirubin and FA is somewhat lower than that of human serum albumin (HSA), their binding positions in space exhibit striking similarity[24]. Furthermore, AFP has a higher affinity for $Zn^{2+}$ than HSA[25]. Protein structure serves as the basis for its functionality, however, detailed reports on the structural characteristics of AFP are currently lacking, making it crucial to analyze its protein structure to gain a comprehensive understanding of its function.

HSA is the most abundant protein in the plasma, with a half-life of 19 days. Both HSA and AFP belong to the albumin family, and HSA is primarily synthesized in the adult liver, with higher synthesis proportions of AFP in the yolk sac and fetal liver[26]. HSA functions include maintaining colloid osmotic pressure in blood vessels, regulating blood pH, acting as a scavenger of free radicals under inflammatory conditions, and participating in processes such as coagulation and wound healing[27]. HSA also plays a crucial role in transporting various bioactive molecules including proteins, FA, hormones, drugs, and nutrients[28,29]. These functions depend on the structural characteristics of HSA, with its internal binding sites and surface-active groups allowing interactions with many biomolecules and transport them throughout the body. Because of its advantageous properties such as high serum concentration, long half-life, frequent recirculation, non-toxicity, and non-immunogenicity, HSA is widely employed as a delivery vehicle for various therapeutic molecules[30,31]. In vivo, AFP exhibits transport functions similar to HSA, including the transportation of maternal nutrients, such as docosahexaenoic acid to the fetus via the placental barrier[32], indicating its potential as a drug delivery vehicle. Owing to the presence of specific AFP-receptors in certain cancer cells[33,34], AFP has shown more

promising potential for targeted drug delivery in cancer treatment than HSA. Although there have been numerous studies on the combination of AFP and drugs for tumor treatment, the binding mechanism and mode of action of AFP with drugs have scarcely been addressed. The development of HSA-based drug delivery systems provides guidance for AFP detection. However, the successful development of AFP and AFP-binding drugs relies on a comprehensive understanding of protein structure. Therefore, there is an urgent need to elucidate the structure of AFP.

In this study, we harnessed the HEK 293F cell line to produce recombinant AFP and successfully elucidated its structure using single-particle cryo-electron microscopy(cryo-EM). This study presents a detailed analysis of its structural characteristics, including N-glycosylation, fatty acids, and metal ion binding sites, and a systematic comparison with HSA.

## Results
### The overall structural characteristics of AFP
To study the structural characteristics of AFP, recombinant AFP with a C-terminal 2× Strep tag was transiently expressed in HEK 293F suspension cells. We observed that AFP eluted as a single and symmetric peak and was highly pure by size-exclusion chromatography and sodium dodecyl sulfate-polyacrylamide gel electrophoresis (SDS-PAGE) (Fig. 1a, b). Western blotting confirmed the identity of these proteins (Fig. 1c). These results indicated that we obtained high-quality recombinant AFP for cryo-EM analysis.

We recorded and processed videos of AFP in different states using cryo-SPARC (Fig. 2a, b). After motion correction and contrast transfer function estimation, cryo-EM maps were generated at an overall resolutions of 3.31 Å according to the gold-standard Fourier shell correlation of 0.143 (Fig. 2d and Supplementary Fig. 1), which showed an excellent side-chain density (Fig. 2c). This allowed us to build accurate atomic models for most regions of the AFP (Fig. 2e).

The AFP had a heart-like asymmetric shape with dimensions of 93.2 Å × 83.3 Å × 59.6 Å and volume of 36,030 Å³. Full-length AFP exhibited a signal peptide (residues 1–18) and mature protein (residues 19–609)[35] which were observed in the three-dimensional structure (Fig. 2e). AFP predominantly consists of α-helices accounting for approximately 69% of its secondary structure, which are connected by loop regions (Fig. 2e and Supplementary Fig. 2). AFP has three domains: I, II, and III, which are further subdivided into two subdomains: IA, IB, IIA, IIB, IIIA, and IIIB. These subdomains consisted of 4–6 helices and give rise to potential ligand-binding sites (Fig. 2e). We identified N-glycosylation at the N251 position of AFP as well as the binding of metal ions. Additionally, we noted the presence of four distinct natural fatty acid-binding sites within AFP, which are situated in subdomains IIA, IIA/IIB, IIIA, and IIIB (Fig. 2e). These findings contribute to our understanding of the structural characteristics of AFP.

### The glycosylation characteristics and a potential metal ion binding site in AFP
After a thorough investigation of the AFP structure, an additional electron density map was detected in the Asn251 of AFP. N-linked glycosylation is a co-translational or post-translational modification in which a sugar chain is attached to a specific asparagine residue with the consensus amino acid sequence N-X-S/T, where X represents any amino acid except proline. The amino acid sequence surrounding N251 in AFP was TKVNFTEIQ, which conforms to the criteria for N-linked glycosylation (Fig. 3a). Therefore, we preliminarily conclude that this density is the structure of the sugar chains. Through the introduction of a mutation, substituting asparagine with serine (N251S), we observed a higher electrophoretic mobility of the mutant AFP by SDS-PAGE (Fig. 3b), indicating that the mutation reduced the molecular weight of AFP due to the loss of glycosylation. This further confirmed that the electron density map was a glycosylation modification. Subsequent LC-MS/MS analysis confirmed the N-linked glycosylation of AFP at residue N251, which exhibited diverse sugar compositions. The main sugar composition identified for wild-type AFP was HexNAc (5) Hex (4) Fuc (1), accounting for 26.82% (Supplementary Table 1). N-glycosylated glycan

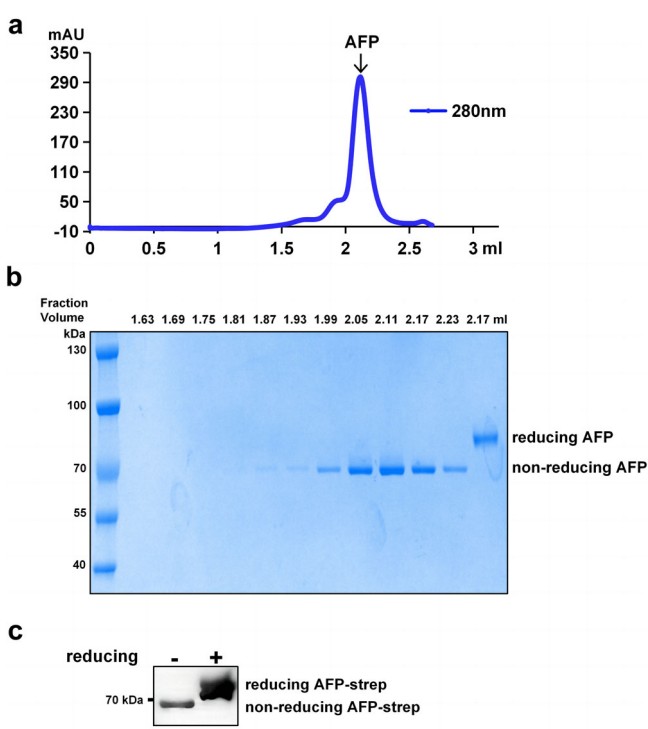

**Fig. 1 | Extraction and purification of AFP. a** Size-exclusion chromatography profile of AFP. The fraction volume of peak was 2.12 mL. **b** SDS-PAGE analysis of purified AFP from the size-exclusion chromatography fractions. The molecular weight of the reduced AFP was approximately 85 kDa when using protein loading containing the reducing agent beta-mercaptoethanol. The molecular weight of the non-reduced AFP was approximately 70 kDa when using protein loading without the reducing agent. **c** Western blot analysis of the purified AFP. Immunostaining was carried out with a anti-strep antibody.

**Fig. 2 | Cryo-EM analysis of AFP. a** A raw micrograph of the frozen-hydrated AFP taken from the 300 kV Titan Krios microscope. **b** Typical 2D class averages of AFP. **c** Selected densities from the AFP map with the corresponding atomic models docked. **d** The final 3D reconstruction map of AFP at 3.31 Å resolution. **e** The overall cryo-EM structure of AFP.

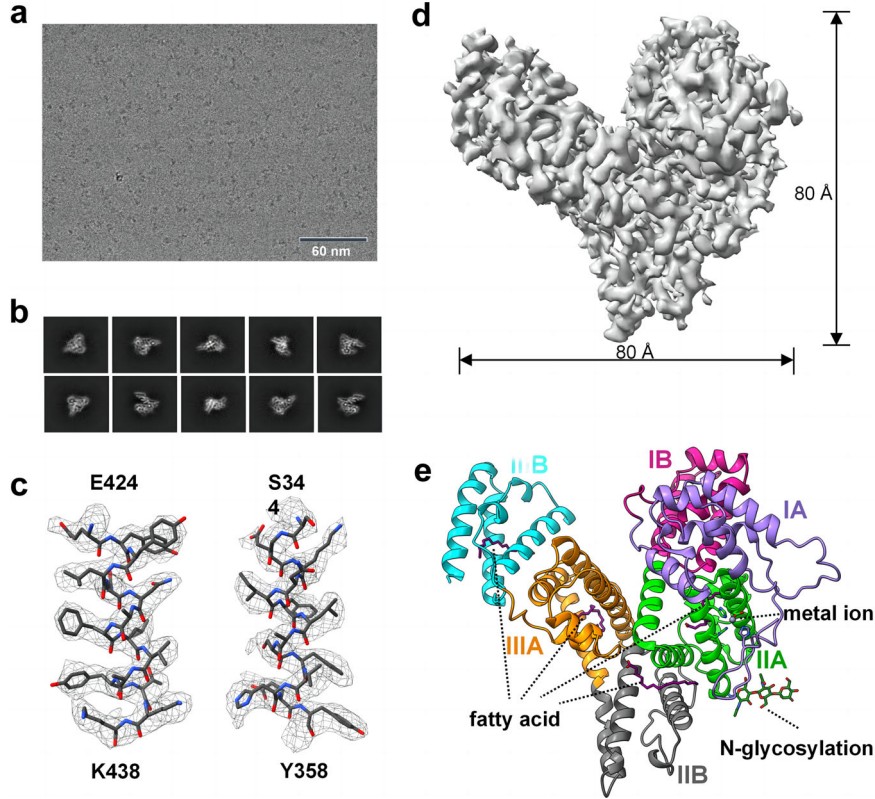

**Fig. 3 | N-glycosylation of AFP. a** Additional electron density map linking to the Asn251 of AFP. The density map is color-coded according to AFP subdomain, and the map of the extra electron density map is marked in red. AFP sequences conform to the N-linked glycosylation sequence pattern. **b** SDS-PAGE analysis of AFP and mutant AFP (N251S). **c** Oligosaccharide structure of AFP. The oligosaccharide is called beta - D - mannopyranose - (1-4) - 2 - acetamido - 2 - deoxy - beta - D - glucopyranose - (1-4) - 2 - acetamido - 2 - deoxy - beta - D - glucopyranose.

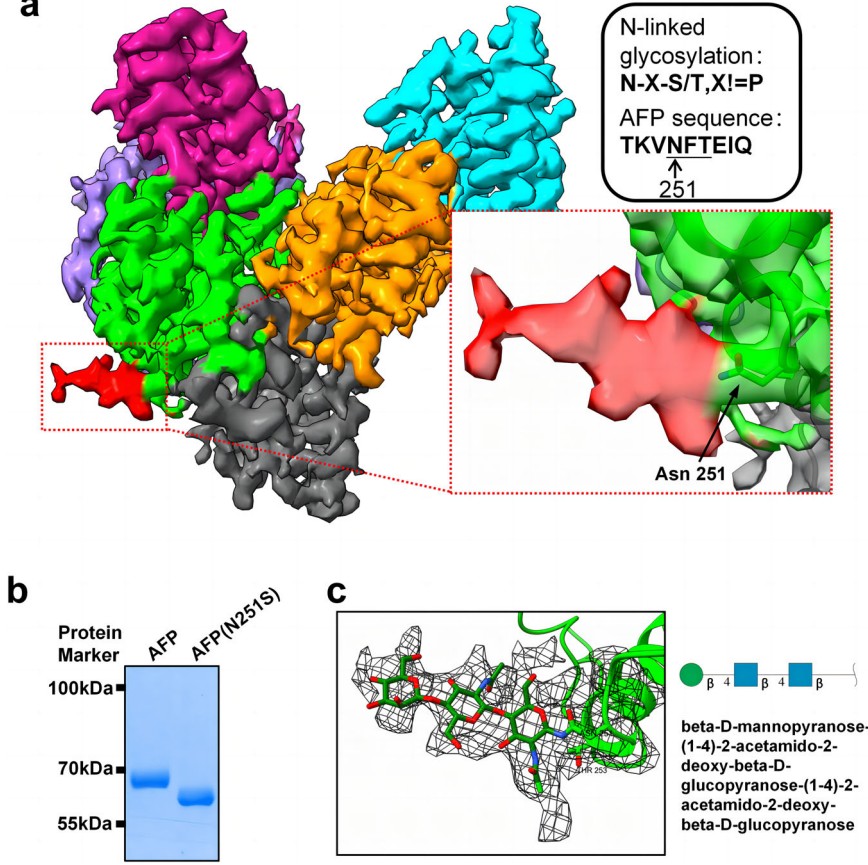

chains generally have a core pentasaccharide structure consisting of two N-acetylglucosamines and three mannose residues[36]. We found that the electron density observed could accommodate three monosaccharides, which corresponded to two 2-acetamido-2-deoxy-beta-D-glucopyranose units and one beta-D-mannopyranose unit, whereas the remaining two β-D-mannopyranose units exhibited less distinct electron densities because of their inherent flexibility (Fig. 3c). In summary, our findings highlight the presence of N-linked glycosylation in AFP and provide partial insights into sugar chain structures. The diverse compositions of N-glycans can potentially influence the biological functions of AFP.

Furthermore, we observed a distinct region of electron density between the side chains of the four amino acids H22, H264, H268, and D280. Based on the known ability of AFP to bind certain metal ions[37,38] and the identification of a similar motif in the Cu, Zn-SOD structure (PDB:5k02) that facilitates zinc ion binding (Supplementary Fig. 4b), it is preliminary to conclude that this electron density corresponds to a metal ion. We mutated four amino acids of AFP to alanine, obtaining the mutant AFP-4mut (H22A, H264A, H268A, and D280A). Following this, we assessed the binding of metal ions to both the wild-type AFP and the mutant AFP-4mut protein using inductively coupled plasma-mass spectrometry (ICP-MS). The findings indicated a reduction in the molar ratio of metal to protein in the mutant AFP-4mut, specifically for elements such as magnesium, aluminum, nickel and zinc (Supplementary Fig. 4a). This implies that the identified four amino acids in AFP possess the capacity to bind to different types of metal ions. Moreover, the molar ratio of some metal elements with AFP exceeds 1, it suggests the potential for AFP to bind to metal ions at multiple locations. After we selected $Zn^{2+}$ fit into the density map (Fig. 4a), it showed a tetrahedral coordination with these four amino acids (Supplementary Fig. 4b). We analyzed the bond lengths and metal coordination via the CheckMyMetal server (Supplementary Fig. 4c). The average distances of the zinc ion to three nitrogen atoms and one oxygen atom was 2.35 Å and 2.45 Å, respectively. Some parameters, such as Valence and nVECSUM, were not ideal. This could be due to the presence of multiple metal ions at this binding site, and the metal ion density being obtained through an averaging algorithm, leading to some deviation when a single metal element is fitted into this density. This motif provides AFP with strong metal-ion binding capabilities. The coordination of $Zn^{2+}$ by H22 further stabilizes the N-terminal loop structure of AFP. Comparing the structures of HSA-$Zn^{2+}$ (PDB:5IJF) and AFP, we noted disparities in the spatial arrangement and quantity of amino acids involved in zinc ion binding (Fig. 4b). Among the amino acids involved in $Zn^{2+}$ binding, HSA lacks histidine and its role is replaced by a water molecule. We compared the metal ion binding motifs of AFP across different species and found differences in sequence conservation between viviparous and oviparous animals.(Fig. 4c). These findings suggest a potential evolutionary divergence in the functional role of AFP in metal ion binding. It is plausible that these differences are associated with differences in reproductive strategies.

### Characteristics of AFP binding fatty acids

In the AFP model, certain regions exhibited distinct density maps, suggesting independently bound ligands. Based on previous research on the binding of AFP to fatty acids[39,40] and the unique characteristics of these irregular density strips, it is tentatively speculated that these are the naturally bound fatty acids of AFP. According the results of GC-MS, it was identified that palmitic acid (C16:0) was the most abundant FA bound to AFP (Fig. 5a), accounting for 57.42% of the total FA. The size of the palmitic acid was appropriate for these densities (Fig. 5b). These FA binding sites are located in AFP substructures IIA, IIA/IIB, IIIA, and IIIB, respectively, and are named FA binding site-1/2/3/4 according to the amino acid sequence order.

The binding characteristics of FA in AFP are as follows (Fig. 5b): At FA binding site-1, the FA interacts with the IIA substructure of AFP. Five helical amino acid residues are mainly involved in the interaction with FA, and the FA chain is mostly surrounded by hydrophobic amino acids. The orientation of the carboxyl group of FA is determined by the positively charged

polar amino acids Lys242 and Lys246, whereas His266 and Met285 contribute to the formation of the S-shaped FA conformation. FA binding site-2 is located at the interface between AFP substructures IIA and IIB, within a shallow groove formed by four alpha helices. Three amino acid residues, Arg233, Asn348, and Gln378, act as anchor points to secure FA at this binding site. The orientation of the FA carboxylate head group was determined based on the HSA-FA binding mode. At FA binding site-3, the FA adopted a U-shaped bend, completely occupying the binding pocket within substructure IIIA of AFP. Below the FA, Hydrogen bonds are formed between Glu474 and Arg372, Ser408, and Arg500, forming a semi-closed pocket with other amino acids, which may be responsible for the FA chain's failure to extend downward. The carboxylic acid groups of FA may interact with Tyr435, Lys438, and Ser513 residues. FA binding site-4 is formed by a hydrophobic channel that spans the width of the subdomain IIIB. The FA extended linearly within the binding pocket. Several polar amino acids, including Tyr425, Tyr426, Asn429, and Lys549 determine the position of the FA carboxyl group.

Analysis of the hydrophobicity of AFP revealed that the overall external surface of the protein is hydrophilic, favoring its presence in the bloodstream. Conversely, FA-binding pockets are formed by hydrophobic surfaces, making them suitable for interaction with hydrophobic molecules such as FA (Supplementary Fig. 3a). Additionally, AFP exhibits a higher proportion of negative charges on its surface, which contributes to its lower isoelectric point. Notably, most of the positively charged amino acid side chains within the entrance of the FA-binding pockets, particularly arginine and lysine, likely aided in anchoring the negatively charged carboxyl group of FA (Supplementary Fig. 3b). Furthermore, we tested the thermal stability of the protein using NanoTemper Tycho NT.6, and found that the thermal stability of the defatted AFP decreased. However, it recovered its original thermal stability when FA was reintroduced (Fig. 5c), indicating that FA binding promoted the thermal stability of AFP.

### Structural comparison between AFP and HSA

The amino acid sequences of HSA and AFP are highly homologous, and their overall structures are very similar. When comparing the overall structures of AFP and HSA-palmitic acid (PDB:1E7H), domain II was almost completely overlapped. The RMSD of HSA and AFP protein is 1.186 Å between 249 pruned atom pairs (across all 571 pairs: 5.763 Å). Compared to AFP, domain I of HSA was closer to the center of the protein, and subdomains IIIA/B were also closer to each other (Fig. 6a). Similar to HSA, AFP exhibits a large number of disulfide bonds that confer structural stability. AFP possesses 32 cysteine residues, all of which participate in the formation of 16 disulfide bonds. Comparatively, AFP has three fewer cysteine residues and one fewer disulfide bond than HSA. Interestingly, except for the first disulfide bond, the remaining disulfide bond positions corresponded to those of AFP and HSA (Fig. 6a).

We individually compared the subdomains of AFP and HSA, and found that the conformation of each subdomain was almost identical (except for the N-terminal region of AFP). The overall conformational differences were likely caused by the binding of ligands or substrates. Comparing the FA binding sites of AFP and HSA, we found that HSA bound to a higher number of FA than AFP, and some binding sites overlapped between the two proteins (Fig. 6b). Comparing the four FA binding sites of AFP with the corresponding positions in HSA, the following observations were made (Fig. 6c): First, AFP's FA binding sites 2 and 4 of AFP are structurally similar to HSA's FA binding sites 6 and 5 of HSA. The properties of the amino acid residues involved in FA binding are also similar. FA binding site 1 of AFP corresponds to the HSA binding site of HSA 7. Both FA binding pockets exhibited similar conformations. However, there is an additional FA binding at the interface between HSA IIA and IA, and the FA carboxyl group is anchored in HSA subdomain IIA through hydrogen bonds with Tyr150, Arg257, and Ser287 side chains. In AFP, Arg257 and Ser287 are replaced by the nonpolar amino acids Gly281 and Gly311, which makes it easier for the FA to bind to AFP IIA without crossing IIA and IA. AFP's FA binding site 3 of AFP corresponds to HSA binding sites 3 and 4.

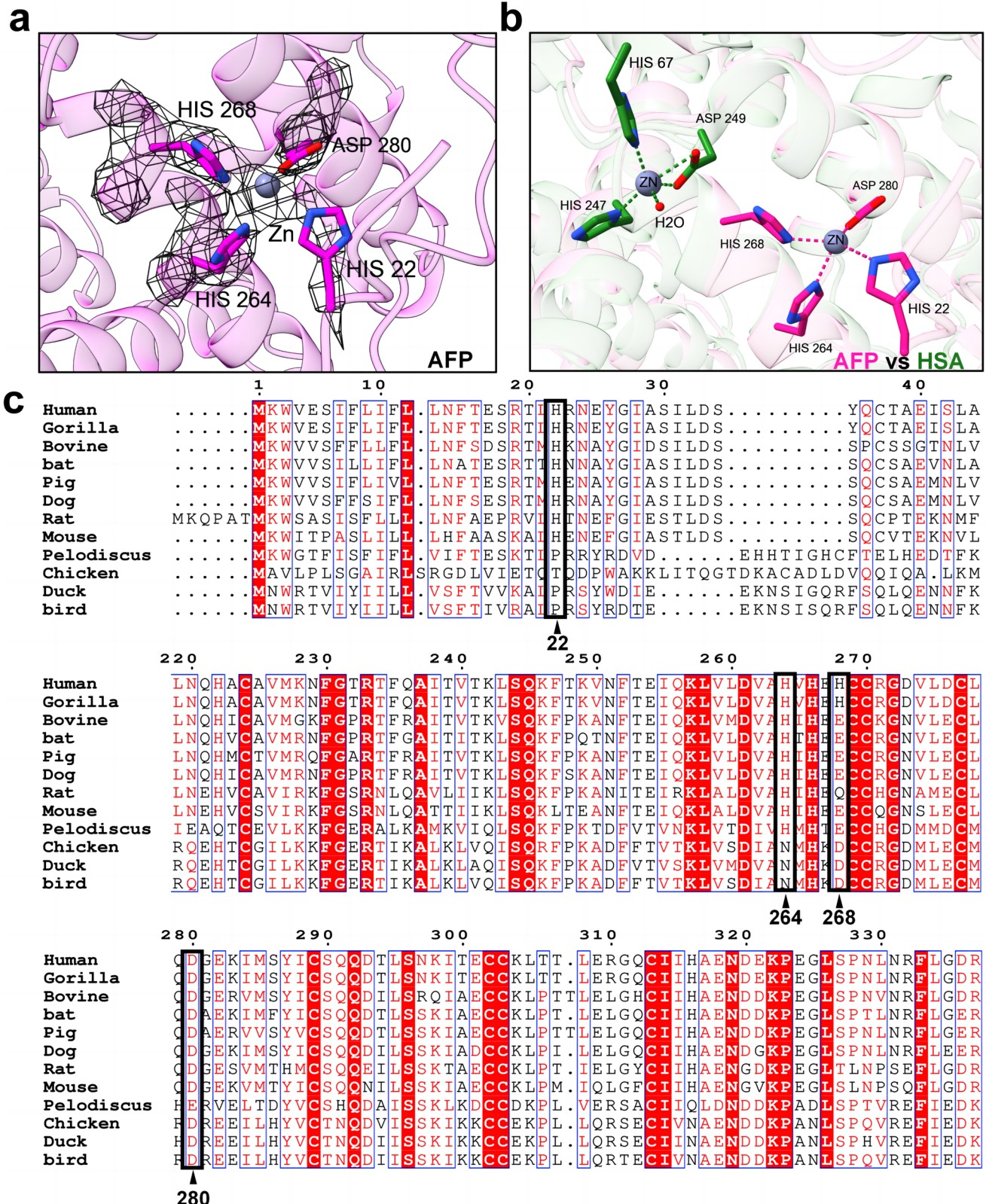

**Fig. 4 | Analysis of AFP-metal ion binding site. a** Atomic models and cryo-EM densities of $Zn^{2+}$ and $Zn^{2+}$ binding site of AFP. $Zn^{2+}$-binding residues are shown as the magenta sticks and $Zn^{2+}$ is shown as the gray sphere. The densities are shown as black mesh. **b** Structural comparison of $Zn^{2+}$ binding site in AFP and HSA. AFP (magenta) and HSA-$Zn^{2+}$ complex (5IJF, lime) are superimposed with the domain I aligned. Zinc ions are shown in gray, oxygen in red, nitrogen in dark blue. **c** Multiple sequence alignment of the residues involved in binding metal ion. The metal ion-binding sites are indicated by black triangles. Invariant and highly conserved residues are shaded red and colored red, respectively.

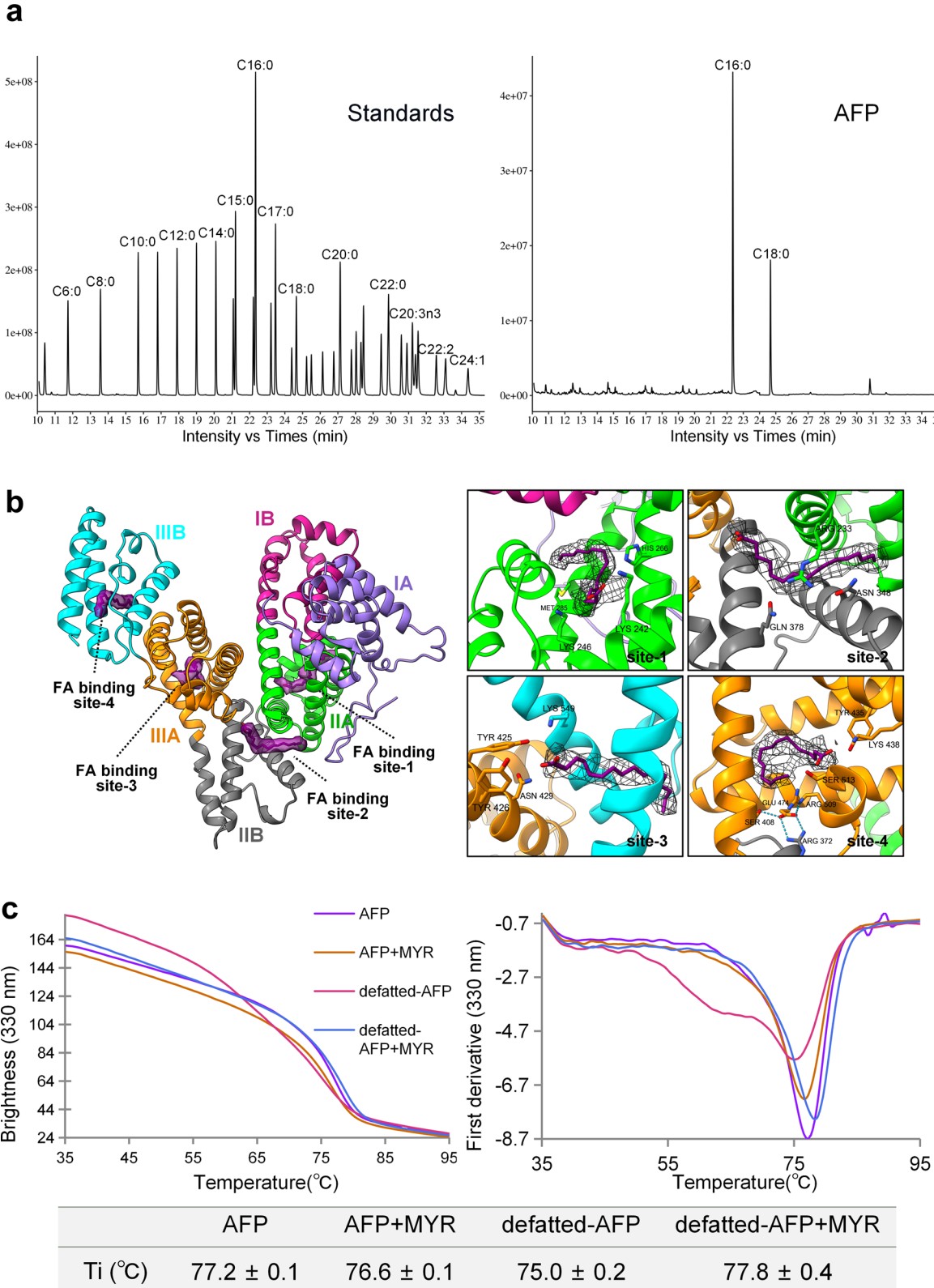

**Fig. 5 | Fatty acids binding sites of AFP. a** GC-MS analysis of fatty acids extracted from AFP. The left panel displays the chromatogram of fatty acid standards. The right panel exhibits the chromatogram of fatty acids extracted from AFP, with the major peaks being identified as palmitic acid (C16:0) and stearic acid (C18:0). All the ions were confirmed by tandem MS spectra. **b** The overall and detail characteristic of fatty acids (FA) binding sites in AFP. The left picture illustrates the spatial positioning of the density map and atomic structure of FA within the AFP structure. The right picture shows the amino acids associated with FA. The FA are palmitic acid (C16:0) and are colored in purple. **c** The thermal stability of AFP was detected by label-free thermal shift assay. The left is the melting profile of AFP and defatted-AFP derived from the 330 nm fluorescence emission in the absence and presence of given myristic acid (MYR), and the right is the first derivative of these traces. The inflection temperatures (Ti) are presented as the mean ± SE of three independent measurements.

**Fig. 6 | Comparison of overall structure and fatty acids binding pockets in AFP and HSA.**
**a** Structural comparison of AFP (8X1N, magenta) and HSA (1E7H, lime) by global alignment. The lime arrows show the direction in which the HSA is offset relative to the AFP. Red dotted circle show HSA' the extra disulfide bond compared with AFP. Blue dotted circle show the non-corresponding disulfide bond between AFP and HSA. **b** Structure of AFP-FA (pink) and HSA-FA (green). The FA are palmitic acid (C16:0), which are colored with magenta in AFP and forest green in HSA, respectively. The different FA binding sites are shown by arabic numerals. **c** Structural comparison of subdomains in AFP (8X1N, magenta) and HSA (1E7H, lime). The FA are colored with purple in AFP and forest green in HSA, respectively. Amino acid residues associated with FA are shown.

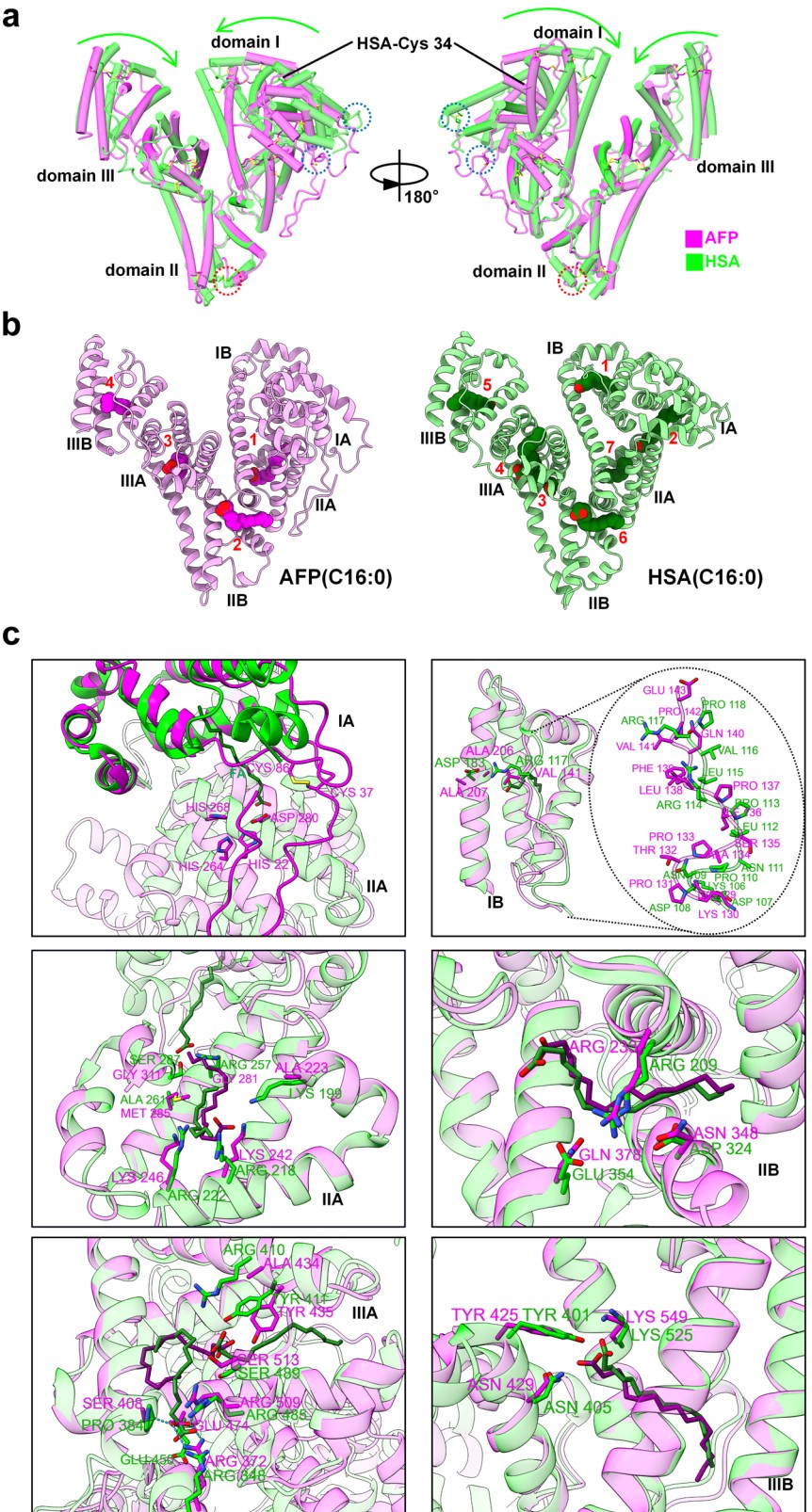

The peptide chain skeleton conformations involved in the binding pocket of both proteins were similar; however, the sizes of the pockets were different. HSA binds to an additional FA compared with AFP in subdomain IIIA. The amino acid residues involved in the interactions with the carboxyl groups of FA are different. In HSA, FA3 forms hydrogen bonds with Ser342, Arg348, and Arg485, whereas FA4 forms hydrogen bonds with Arg410, Tyr411, and Ser489. In AFP, the FA carboxyl group is in close proximity to Tyr435 and Ser513; Glu474 forms hydrogen bonds with Arg372, Ser408, and Arg509, resulting in stabilized and restricted motion of the Glu474 side chain. In HSA, Pro384 corresponds to Ser408 in AFP and leads to the inability to establish a hydrogen bond with Glu450 (corresponding to Glu474 in AFP). In the presence of FA, the Glu450 side chain of HSA underwent outward

conformational fluctuations. This structural variation between AFP and HSA contributes to the comparatively smaller binding pocket size of AFP. Consequently, FA occupies the majority of the binding pocket in AFP, thereby impeding its ability to bind excessive amounts of FA.

Unlike HSA, the IB subdomain of AFP did not bind to FA. There were striking differences in the amino acid properties involved in FA binding between HSA and AFP, such as Arg117 (HSA), corresponding to Val141 (AFP), and Asp183 (HSA), corresponding to Ala206/207 (AFP). We compared differences in this subdomain between the two proteins and found changes in the amino acid composition of the loop. AFP has two additional amino acids in the loop compared to HSA. The two positively charged arginine residues (Arg114/Arg117) in the HSA loop may play an important role. In AFP, these residues corresponded to nonpolar amino acids (Leu138/Val141) (Fig. 6c), affecting the surface charge and hydrophobicity of the binding pocket. Therefore, loop differences may be a determining factor for substrate-binding in this pocket for both proteins.

We performed a comparative analysis of all subdomains of AFP and HSA, and found that almost all subdomains exhibited a similar conformation between the two proteins. A major conformational difference was observed in the IA subdomain, which was primarily concentrated in the N-terminal region. The 24 N-terminal amino acids (AA 19–42) in AFP form a loop, where Cys37 and Cys86 from another loop form a disulfide bond (Fig. 6c), stabilizing the conformation of both loops. Furthermore, AFP' His22, along with three amino acids from domain IIA, forms a tetrahedral coordination with a metal ion, further enhancing the stability of the loop structure. In contrast, the N-terminal amino acids in HSA form an α-helix that, along with other helices, creates a small hydrophobic pocket. The HSA FA2 chain tail binds to this pocket with its carboxyl group anchored to domain IIA (Fig. 6c). Consequently, FA binding to HSA domain I enabled the rotation of domain I relative to domain II, thereby facilitating the formation of a continuous hydrophobic pocket. This is one of the reasons why HSA exhibited a more compact overall conformation.

## Discussion

Currently, research on AFP has mainly focused on its role as a tumor/pregnancy marker and its cellular biological functions, whereas studies on the protein structure and properties of AFP have received less attention in recent years. The direct observation of AFP structure dates back to 1983, Luft, et al. acquired the 'V-shaped' images of human AFP molecules by combining dark-field electron microscopy with a laser-assisted optical system[41]. Techniques such as circular dichroism, fluorescence spectroscopy, and scanning microcalorimetry have been used to study the secondary structure and conformation of AFP, as well as its protein properties such as stability and ligand binding. Vladimir N. Uversky's experimental findings demonstrated that AFP exhibits high protein stability, maintaining a rigid native structure under various conditions (e.g., pH 4.5–10.5; below 70 °C; below 2.0 M guanidinium hydrochloride or 7.5 M urea)[42,43]. Although a 3D model of AFP has been simulated and utilized[44,45], the true conformation of AFP remains elusive, impeding a deeper understanding of its structural properties. In our previous study, we used single-particle cryo-EM to elucidate the structure of AFP[46]. Unlike X-ray crystallography, cryo-EM does not require sample crystallization, which is a prominent advantage for large and complex molecules that are difficult to crystallize. Our successful resolution of the approximately 69 kDa small protein using cryo-EM represents a breakthrough in its application. Elucidation of the structure of AFP is highly valuable for studying its functions and applications. We conducted a detailed analysis of the natural structural features of AFP, including N-glycosylation modifications, naturally bound metal ions, and FA, and similarities and differences with the HSA structure.

AFP is a glycoprotein that can be classified into AFP-L1, AFP-L2, and AFP-L3, based on its affinity for lens culinaris agglutinin. AFP-L3 has the highest affinity for lens culinaris agglutinin and is a new-generation tumor biomarker that distinguishes itself from total AFP[19]. AFP-L3 is characterized by an additional α-1,6-fucosylated residue at the reducing end of N-acetylglucosamine in its glycan chain. We performed glycosylation mass spectrometry analysis on AFP extracted from normal HEK 293F cells and liver cancer cells Bel7042 (293-AFP and 7402-AFP). The results showed that the glycan composition of AFP derived from both cell lines contained fucosylated glycans, with the percentage of fucosylated glycans exceeding 75% of all glycans. This indicates that the AFP derived from both cell lines are biased toward AFP-L3. However, their major glycan compositions were different. The major glycan composition of 293-AFP was HexNAc (5) Hex (4) Fuc (1), accounting for 26.82%, while 7402-AFP was HexNAc (4) Hex (5) Fuc (1), accounting for 44.44% (Supplementary Table 1). The differences in the glycan composition of AFP may ultimately contribute to functional differences. Tumor-derived AFP (tAFP), rather than cord natural-derived AFP (nAFP), can directly induce apoptosis in NK cells and DC in vitro[10]. The main difference in glycosylation between tAFP and nAFP is whether the core polysaccharide is linked to fucose residues. However, not all nAFP lack fucose residues, so the reason for inducing immune cell apoptosis may not be solely the presence of fucose residues, but also differences in glycan composition. Further research is required to confirm these findings.

Metal ions play vital roles in biological systems by participating in various processes such as redox reactions, electron transfer, enzymatic catalysis, and protein structure stabilization. HSA is the primary protein responsible for binding and transporting metal ions in the human plasma and is crucial for maintaining the homeostasis of metal ions. In AFP domain IA, we discovered a metal ion-binding site involving the coordination of three His residues and one Asp residue (Fig. 4a). Interestingly, the presence of metal ions was not intentionally introduced during the protein extraction process. However, the electron density map indicated that AFP binds to the metal ions present in the culture medium. Previous studies have demonstrated that AFP exhibits a higher affinity for $Zn^{2+}$ than HSA[25], which can be attributed to structural differences, including the presence of an additional histidine residue involved in binding. AFP serves as the primary transport carrier during pregnancy. One possible explanation for the high affinity of AFP towards metal ions compared to HSA is its requirement to traverse the embryonic-fetal barrier. Additionally, from an evolutionary standpoint, the metal-ion-binding motifs of AFP exhibited greater conservation among the viviparous animals (Fig. 4c). In contrast to oviparous animals, the fetuses of viviparous animals require crossing the placental barrier to obtain essential nutrients from the maternal body, consequently necessitating a stronger affinity. Additionally, there have been studies indicated that AFP shares similarities with Cu,Zn-superoxide dismutase (Cu, Zn-SOD) in its ability to bind copper ions and display superoxide dismutase activity[47]. An intriguingly similar pattern of metal ion binding was observed in the structure of AFP, which aligns with that seen in Cu, Zn-SOD (PDB: 5K02) (Supplementary Fig. 4b). This provides further corroborative evidence that AFP not only facilitates metal ion transport but also possibly implicated in modulating the balance of redox reactions within the bloodstream. Such observations indicate a multifunctional role for AFP, which goes beyond the traditional understanding of its physiological responsibilities.

It is well known that AFP has transport functions, allowing it to transport certain nutrients (FA and metal ions) from maternal blood to embryonic cells through the placenta. We observed the structure of AFP binding to natural FA and analyzed its structural characteristics. These FA binding pockets may also function as potential binding sites for other small-molecule ligands. AFP has specific receptors on its cell surface, including embryonic cells, tumor cells (HCC, breast carcinoma, gastric cancer, etc.), proliferating liver cells, and some immune cells (such as myeloid-derived suppressor cells)[48,49], which consume AFP through its receptor-mediated cell endocytosis[50]. Normal cells in vivo generally have no AFP receptors; therefore, using AFP as a transport carrier can target drug transport to the tumor site without damaging the normal cells. With the increasing development of new treatments for tumors based on AFP characteristics, in patent US20080318840A1, Pak demonstrated that AFP, after co-incubation with Atractyloside, Thapsigargin and Betulinic Acid, had a remarkable therapeutic effect on tumor cells, tumor-bearing mice, and cancer patients. Furthermore, the combination of AFP and 1'-S-1'-acetoxychavicol acetate showed higher efficacy against AFP receptor-positive tumor cells than

treatment alone[51]. Some drugs can exhibit increased antitumor activity after covalent coupling with AFP[52,53]. ACT-903, an anti-tumor drug developed by Alpha Cancer Technologies Inc., is a conjugate of AFP and maytansine. It has been demonstrated to effectively improve the survival of xenograft mice without obvious signs of toxicity[54]. Although there have been numerous studies on the combination of AFP and drugs for tumor treatment, the binding mechanism and mode of action of AFP with drugs have scarcely been addressed. Structural information regarding AFP-FA can provide valuable insights into the development of drugs that bind to AFP. Computer-aided docking technology can be used to screen drugs that have a high affinity for the AFP-binding pocket, leading to potential new strategies for cancer therapy by exploiting the AFP tumor-targeting properties.

AFP and HSA are members of the albumin family and exhibit obvious sequence homology[55]. Comparative analysis of their overall structures and individual domains revealed striking similarities, suggesting potential functional similarities. AFP and HSA function as natural transport vehicles within the bloodstream, reversibly binding to various endogenous and exogenous molecules for transportation to different tissues. HSA-based drug delivery systems have been extensively studied, and some HSA-based formulations have been approved by the FDA for clinical use, including abraxane (HSA-bound paclitaxel), Vasovist (HSA non-covalently bound to Gd), Victoza (glucagon-like peptide-1-FA derivative non-covalently bound to HSA in vivo), Tresiba (human insulin-FA derivative non-covalently bound to HSA in vivo), and Fyarro (HSA-bound rebeccamycin)[56]. Hence, investigations of AFP could draw upon findings pertaining to HSA. One notable similarity between AFP and HSA is their high affinity for FA. Successful examples, such as Victoza and Tresiba, indicated the feasibility of developing FA-therapeutic molecule complexes that bind to AFP. The structural features of the FA-binding pocket in AFP play a crucial role in the identification of suitable FA therapeutic molecules. For instance, the size of the AFP-binding pocket determines the optimal FA chain length for higher affinity and extended half-life in the body. Moreover, the polarity and charge of amino acid residues around the pocket entrance influences FA binding. Consequently, the incorporation of appropriate modification groups (e.g., the γ-Glu linker in Victoza and Tresiba) can enhance the affinity between AFP and FA. However, AFP and HSA are not identical and HSA possesses more binding sites than AFP, which may allow it to interact with a greater number of biomolecules. This could be one reason why HSA, rather than AFP, is the most abundant transport carrier in the bloodstream of adults. Additionally, the different modes of metal ion binding between AFP and HSA may contribute to their distinct structures and functions. One of the functions of glycosylation modification is to recognize cell surface-specific receptors. HSA lacks glycosylation modifications, whereas AFP is primarily expressed at high levels in embryonic or cancerous tissues, suggesting that AFP may have a preference for recognizing cells with higher stemness, which could be related to its glycosylation modification. Therefore, the characteristic glycosylation modification of AFP may be a reason for its re-expression during tumorigenesis.

This study is the first to report the natural structural characteristics of AFP, including N-glycosylation, metal ion binding, and FA binding, as well as the structural similarities and differences with HSA. AFP plays a role in the occurrence and development of HCC, and elucidation of AFP structure provides insights into its biological functions, such as the cooperative carcinogenic effect of AFP and its ligands (FA and metal ions). Differential glycosylation modifications of AFP in different environments have resulted in the emergence of AFP-L3 as a novel tumor marker. In addition, glycosylation modifications of AFP may play a role in the recognition of specific receptors in certain cells. The structural disparities between AFP and HSA partially contributed to the tissue and temporal specificity of their expression. Moreover, AFP and HSA exhibited numerous similarities, such as overall conformation resemblances and multiple high-affinity FA binding sites, which enabled AFP to possess potential as a drug delivery system that holds great promise for the treatment of certain cancers (e.g., liver cancer). In conclusion, this study provides a foundation for an in-depth investigation of AFP through structural characterization.

## Methods

### Protein expression and purification

The coding region of the full-length *human afp* gene (NM_001134.2) was cloned into the pCAG vector containing a C-terminal tandem 2× Strep tag using homologous recombination with a ClonExpress Ultra One Step Cloning Kit (Vazyme Biotech). The *afp* mutants (N251S and H22A + H264A + H268A + D280A) were obtained by the Fast Mutagenesis System kit (TransGen Biotech). Sequences of the primers used in this study are summarized in Supplementary Table 2. HEK-293F cells were cultured in serum-free SMM 293-TII medium (Sino Biological) at 37 °C under 5% $CO_2$ and centrifuged at 110 rpm. Human AFP was transfected into HEK-293F cells when the cell density reached approximately $2 \times 10^6$–$2.5 \times 10^6$ cells/mL. For transfection, 1.5 mg plasmid and 3.0 mg linear polyethyleneimine (PEI, Polysciences) were separately mixed into 40 mL fresh cell culture medium. The transfection mixture was incubated for 15–25 min at room temperature before being added to 800 mL cells for transfection. The transfected cells were cultured for 48 h before harvesting. After centrifugation at 2000 rpm for 10 min, the cell pellets were resuspended and washed with 1× PBS, quickly frozen in liquid nitrogen, and stored at −80 °C until further use. The pellets were resuspended in lysis buffer consisting of 100 mM Tris-HCl (pH 8.0), 150 mM NaCl, 1% Triton X-100, 5% Glycerol, 1 mM PMSF and Protease Inhibitor Cocktail on ice for 30 min. All subsequent steps were performed at 4 °C. After centrifugation at $20,000 \times g$ for 1 h, the supernatant was mixed with Strep-Tactin® Sepharose® (IBA, LifeSciences) for 2 h at 4 °C. The resin was washed with 10 column volumes of a buffer comprising 100 mM Tris-HCl (pH 8.0), 300 mM NaCl. Recombinant AFP protein was eluted with 50 mM biotin in the presence of 100 mM Tris-HCl (pH 8.0) and 150 mM NaCl. The elution was concentrated and subjected to size-exclusion column chromatography using a Superose 6 Increase column (GE Healthcare) equilibrated with a buffer comprising 25 mM Tris-HCl (pH 8.0) and 150 mM NaCl. The peak fractions were collected and concentrated for cryo-EM sample preparation and further biochemical studies.

### Identification and analysis of glycosylation modification

Liquid Chromatography Tandem Mass Spectrometry(LC-MS/MS) was used to identify glycosylation of the purified AFP protein. First, the protein samples were subjected to SDS-PAGE and the target strips were cut. After decolorization, dehydration, and reduction alkylation, the gel slices were digested with chymotrypsin and the peptides were extracted and lyophilized. The peptides in 10 μL of 0.1% formic acid before LC-MS/MS analysis. The sample was injected into an LC-MS/MS system (Easy-nLC 1000 liquid chromatograph and Orbitrap 240 Mass Spectrometer, Thermo Scientific) equipped with an Acclaim PepMap RPLC C18 column (1.9 μm, 150 μm internal diameter ×150 mm, Thermo Scientific). Mobile phases A and B were 0.1% formic acid in water and 80% acetonitrile/0.1% formic acid in water, respectively. The total flow rate was 600 nl/min. LC linear gradient: from 4% to 8% B for 2 min, from 8% to 28% B for 43 min, from 28% to 40% B for 10 min, from 40% to 95% B for 1 min, and from 95% to 95% B for 10 min.The mass spectrometer was operated with a spray voltage of 2.2 kV. Full-scan mass spectrometry data were obtained in the m/z range 300–1800. The MS/MS parameters included the HCD activation type, normalized collision energy of 28.0, activation time of 66.000, and up to the top 20 most intense peptide ions from the preview scan using the Orbitrap instrument.

Raw Mass Spectrometry(MS) files were analyzed and searched against a target protein database based on sample species using Byonic. The parameters were set as follows: the protein modifications were carbamidomethylation (C) (fixed), oxidation (M) (variable), acetyl (Protein N-term) (variable), glycan 309 mammalian no sodium.txt @ NGlycan (variable), glycans 78 mammalia.txt @ OGlycan (variable), enzyme specificity was set to chymotrypsin, maximum missed cleavages were set to 3, precursor ion mass tolerance was set to 20 ppm, and MS/MS tolerance was 0.02 Da. Only highly confidently identified peptides were selected for downstream protein identification analysis.

## Identification and analysis of metal ions

In order to analyze the binding of metal ions to AFP, we conducted an assessment of metal ion content before and after AFP mutation using Inductively coupled plasma mass spectrometry (ICP-MS). Firstly, wild-type AFP and mutant AFP-4mut (H22A, H264A, H268A, and D280A) were extracted and purified. The protein concentrations were determined using the BCA method, following which weighed amounts of wild-type AFP and mutant AFP-4mut protein solutions, 0.1876 g and 0.1747 g respectively (at a protein concentration of 1 mg/mL), were placed in cleaned polytetra-fluoroethylene (PTFE) digestion vessels. Subsequently, 1 mL of concentrated nitric acid was added, the vessels were sealed, and a microwave digestion program was run: maintaining pressure at 38 MPa, temperature at 190 °C, with power held at 800 W for 5 min, followed by 1400 W for 20 min, and ceasing power for a final 15 min. Upon completion of digestion, the samples were transferred to clean polyethylene terephthalate (PET) plastic bottles and diluted to a known weight of 10.00 g with ultra-pure water. A parallel blank control underwent identical treatment. The samples were then injected into an ICP-MS system (ICAP QC, ThermoFisher Scientific). Instrument detection parameters encompassed a radio frequency power of 1550 W, a cooling gas flow rate of 14 L/min, an auxiliary gas flow rate of 0.7763 L/min, a nebulizer gas flow rate of 1.0749 L/min, a nebulizer temperature of 2.5 °C, a peristaltic pump rate of 40 r/min, a dwell time of 0.1 s, 3 acquisition cycles, and KED detection mode. The ICAP QC software autonomously collected and analyzed the obtained data.

## Identification and analysis of fatty acids

To extract fatty acids, chloroform and methanol were mixed with 0.5 mL of AFP (0.76 mg) using a volume ratio of 1:2:2. The mixture was sonicated for 10 min in an ice-water bath before being spun at 2500 g for 10 min. The chloroform phase was collected from the bottom. After drying under a nitrogen stream, 2 mL of methanol (containing 5% sulfuric acid) was added, and the mixture was placed in a water bath at 80 °C for 2 h. After cooling, 2 mL of n-hexane and 1 mL of water were added, and the mixture was vortexed for 30 s. The supernatant was collected after centrifugation at 2000 rpm for 5 min and dried under a nitrogen stream. An appropriate volume of isooctane was added based on the sample concentration used for detection.

Gaschromatography-mass spectrometry (GC-MS) was used to further identify the endogenous fatty acids bound to the purified AFP protein. The sample was injected into an GC-MS system (Agilent 7820 gas chromatograph and Agilent 5977 mass spectrometer, Agilent Technologies) equipped with an CP-Sil 88 gas chromatographic column (100 m × 0.25 mm × 0.25 µm, Agilent Technologies). The split ratio was 10:1, carrier gas was high-purity helium, and flow rate was 1.0 mL/min. The initial temperature of the column oven was 100 °C for 5.0 min, and the temperature was programmed to 240 °C at 4 °C/min for 15 min. The mass spectrometry analysis conditions were as follows: the inlet temperature was set at 260 °C and the quadrupole temperature was set at 150 °C. The scan mode was set to single ion monitoring (SIM) with a mass scanning range of 30–550 m/z. Data were acquired using MassHunter GC/MS Acquisition (Agilent Technologies) and processed using a Quant-My-Way (Agilent Technologies).

## Preparation of defatted AFP

The recombinant AFP protein was defatted using charcoal The detailed experimental procedure is as follows: AFP (0.8 mg) diluted to 1 mL with PBS was added to 4 mL charcoal (5 mg in PBS) at 0 °C. The pH was then carefully adjusted to 3.1 with 0.1 mol/L HCl, and the mixture incubated with shaking at 0 °C for 3.5 h. The solution was then centrifuged thrice at 4000 rpm for 15 min to remove charcoal. The supernatant containing lipid-free protein was concentrated to 100 µL and subjected to size-exclusion column chromatography using a Superdex® 200 Increase column equilibrated with PBS buffer (pH 7.3). The peak fractions were then collected.

## Label-free thermal shift assay

Label-free thermal shift analysis monitors changes in emission intensity and wavelength maximum of intrinsic fluorescence properties of buried Trp and Tyr residues in proteins that become exposed in the unfolded state when the protein is exposed to a temperature increase from 35 °C to 95 °C. Myristic acid (MYR) was purchased from MedChemExpress (US). Tycho NT.6 instrument was used to perform label-free thermal shift analysis. To this end, recombinant human AFP and defatted-AFP solutions were diluted in PBS to a final concentration of 1 mg/mL with and without 100 µM MYR. Samples were incubated for 20 min at room temperature before loading into Tycho NT.6 capillaries (NanoTemper Technologies, cat#TY-C001) and the thermal unfolding profiles of both proteins in the presence and absence of the substrate. Experiments were performed in triplicate, and the inflection temperature (Ti) values are reported as the mean ± standard deviation.

## Cryo-electron microscopy sample preparation and data collection

Three microliters of human AFP at a concentration of 1 mg/mL was applied to glow-discharged (60 s at 15 mA) 300-mesh Quantifoil R 1.2/1.3 grids, and subsequently blotted using a Vitrobot Mark IV (Thermo Fischer Scientific) at 4 °C and 100% humidity, and then frozen in liquid ethane. The grids were imaged on a Thermo Fisher Krios G3i microscope (Thermo Fisher Scientific) equipped with a GIF Quantum energy filter (Gatan) and K3 Summit detector (Gatan). The energy filter was operated with a slit width of 20 eV to remove the inelastically scattered electrons. Image stacks were collected using EPU2 software at a pixel size of 0.526 Å/pixel with a total dose of 50 e$^-$/Å$^2$ and a defocus range of −1.0 to −2.0 µm. Each stack contains 32 frames. The data collection parameters are summarized in Table 1. The instrument is located at

**Table 1 | Data processing details, refinement, and validation statistics**

| | AFP (EMD-37997, PDB: 8X1N) |
| --- | --- |
| Data collection and processing | |
| Magnification | 165,000× |
| Voltage (kV) | 300 |
| Electron exposure (e$^-$/Å2) | 50 |
| Defocus range (µm) | 1.0–1.6 |
| Pixel size (Å) | 0.526 |
| Symmetry imposed | C1 |
| Micrographs | 10,849 |
| Initial particle images (no.) | 739,577 |
| Final particle images (no.) | 144,221 |
| Resolution offinal map [Å] | 3.31 |
| FSC threshold | 0.143 |
| B-factor applied [Å2] | −149.8 |
| Refinement | |
| Model composition | |
| Non-hydrogen atoms | 4763 |
| Protein residues | 591 |
| R.m.s deviations | |
| Bond lengths (Å) | 0.003 |
| Bond angles () | 0.549 |
| Validation | |
| MolProbity score | 1.15 |
| Clashscore | 2.85 |
| Poor rotamers (%) | 0.00 |
| Ramachandran plot | |
| Favored (%) | 97.62 |
| Allowed (%) | 2.38 |
| Disallowed (%) | 0.00 |

the Cryo-EM Center of the Southern University of Science and Technology in Shenzhen, Guangdong Province, China.

### Cryo-EM data processing, model building and refinement

The collected data were processed using cryoSPARC (v3.3)[57], movies of the single datasets were motion-corrected using Patch Motion Correction, and the contrast transfer function (CTF) was estimated using Patch CTF estimation in cryoSPARC. One thousand images were used to generate an initial particle set using a blob picker in cryoSPARC; the particles were binned to a pixel size of 1.052 Å and extracted with a box size of 192 pixels. Two-dimensional (2D) classification was performed using cryoSPARC. High-quality 2D class averages representing projections in different orientations were selected as templates for Topaz[58] training of the entire dataset. The particles were then subjected to 2D classification using Cryo-SPARC. After ab initio model building and six rounds of heterogeneous refinement in cryoSPARC, most of the bad particles were removed, and the selected particles were used to generate the final map using non-uniform refinement with an estimated average resolution of 3.31 Å, using the gold standard FSC (FSC = 0.143).

As a starting point, the model predicted by AlphaFold2 was manually docked onto a map using UCSF ChimeraX[59] (version 1.4). The presented model was manually refined using COOT[60] (version 0.9.8) and automatically refined with Phenix Real-space refinement tool[61] (Phenix v1.20.1).

### Figure preparation and data analysis

Images of the models and maps were prepared using ChimeraX[62]. Multiple sequence alignment was performed using NCBI Clustal Omega[63] and visualized using ENDscript[64].

### Reporting summary

Further information on research design is available in the Nature Portfolio Reporting Summary linked to this article.

### Data availability

The cryo-EM structures of AFP were deposited in the Protein Data Bank (PDB) under code 8X1N. The cryo-EM density maps of the structure were deposited at the Electron Microscopy Data Bank (EMDB) using the code EMD–37997. Uncropped SDS-PAGE for Fig. 3b shown in Supplementary Fig. 5.

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

## Acknowledgements

We sincerely thank all staff members of the Cryo-EM Centre, the Southern University of Science and Technology (China, Shenzhen) for technical support in cryo-EM data collection, and Dr. Qiujiao Wang and Zijuan Zou for providing the purified proteins. This work was supported by The National Natural Science Foundation of China (Nos. 82060514 and 81960519), the Research Project of Take off the Proclamation and Leadership of the Hainan Medical College (No. JBGS202106), the Natural Science Foundation of Hainan Province (No. 820RC634 and 824RC517), the Hainan Provincial Science and Technology Special Fund (No. ZDYF2021SHFZ222), and Hainan Provincial Association for Science and Technology Program of Youth Science Talent and Academic Innovation (No. QCXM 201922).

## Author contributions

K.L., M.Z., and J.X. designed experiments. K.L., C.W., M.Z., J.X., B.L., and H.L. performed experiments. M.L. and Z.L. designed and supervised the

study and analyzed the data. M.L. wrote the manuscript. All authors contributed to the manuscript and approved the submitted version.

## Competing interests
The authors declare no competing interests.

## Ethics approval
All experiments were approved by the committee of the Hainan Medical College, Haikou, Hainan Province, China.
