## [Peer Review File · Communications Biology]

Structural characteristics of alpha-fetoprotein, including 2 N-glycosylation, metal ion and fatty acid binding sitesReviewers' comments:

Reviewer #1 (Remarks to the Author):

Alpha-fetoprotein is an important macromolecule serving as a tumor marker. Understanding the spatial structure of this protein, leading to determining the detailed mechanism of action of AFP, is crucial to designing new therapeutic strategies, including the use of AFP as a drug carrier.

The authors of the manuscript describe in detail the structural features of recombinant AFP purified from HEK-293F cells, including the overall characteristics, glycosylation, metal binding, and ligands. Still, there is no possibility to verify the structural features since the whole manuscript lacks PDB and EMDB codes for obtained structures. A search in the PDB database showed that there is already deposited and published structure of human alpha-fetoprotein (PDB ID: 7YIM), also obtained by cryo-EM microscopy, with a resolution of 2.60 Å.

The lack of access to structural data is particularly important when assessing the correctness of the metal and ligand binding sites. In the description of the experiment, the authors mention that zinc was not added to the sample and it is not included in the buffers and solutions used.

Q1: Has the zinc content been confirmed in the test sample by additional tests, e.g. by absorption spectroscopy/fluorescence? Did the authors analyze the bond lengths and metal coordination in the obtained structure, e.g. via the CheckMyMetal server?

In general, the manuscript is written in a clear way, with only a few editorial errors. The drawings are clear and aesthetic. Unfortunately, due to the inability to verify the correctness and quality of structural data, I cannot recommend this manuscript for publication.

Reviewer #2 (Remarks to the Author):

Alpha-fetoprotein (AFP) is a protein with crucial roles in fetal development and medical diagnostics. During pregnancy, it is produced by the developing liver and yolk sac of the fetus and can be detected in the mother's bloodstream. AFP serves as a significant diagnostic marker, with elevated levels indicating conditions such as liver cancer, germ cell tumors, and neural tube defects in fetuses. AFP is particularly important for the early detection and monitoring of hepatocellular carcinoma (HCC), the most common type of liver cancer. It is suggested that AFP has immunosuppressive properties, which can inhibit immune cell activity. Thus AFP plays a vital role in fetal development and medical diagnostics, with growing potential as a target for cancer immunotherapies.

Authors present a cryoEM structure of human AFP determined at 3.6Å resolution. Protein expressed and purified from HEK293 F cells. They analyse the glycosylation by MS, and describe the FA and metal ion binding sites. They also make structural comparisons with highly similar HSA proteins.

A higher resolution structure of AFP has already been reported at 2.6Å (PDB ID 7YIM). Although it doesn't have any metal ions or glycans modeled.

Major comments:

There is no novelty in the structure as a higher resolution structure exists. Furthermore, multiple structures of highly similar HSA are also available and well described in the literature.

The authors emphasize the glycosylation analysis of heterologously expressed protein which might have no bearing or similarity with native AFP as the glycan composition could be totally different. This analysis would have made better sense on AFP purified from native sources.

The authors discuss differences in conformation with HSA but again physiological relevance of these differences is not clear. Moreover, the differences could be due to variations in protein preparation/buffer conditions.

Druggability of the FA binding sites is discussed, however, no evidence is provided for the same.

Similarities and differences between the metal ion and FA binding sites between HSA and AFP are discussed. Nevertheless, it's important to note that these findings remain speculative as no mutational analysis has been conducted to confirm their validity.

Minor:

RMSD values of HSA and AFP protein should be provided.

Page 13 line 380 words effective and effect are used in the same sentence. May be modified or replaced with a synonym.

The authors have frequently emphasized the elucidation of the structure of AFP, specifically using the terms "actual structure" or "true structure." It is worth noting that they have determined the structure of a recombinantly expressed protein through the application of single-particle cryo-EM, an averaging method. Therefore, the terms "actual" and "true" should be removed.

Respond to the reviewers' comments

Dear Editor,

We are very grateful to the Reviewers and Editor for their comments on our manuscript. We have revised the manuscript according to the Reviewer's advises. Here are our point-by-point answers to the Reviewers' and editor's comments.

Reviewers' comments:

Reviewer #1 (Remarks to the Author):

Alpha-fetoprotein is an important macromolecule serving as a tumor marker. Understanding the spatial structure of this protein, leading to determining the detailed mechanism of action of AFP, is crucial to designing new therapeutic strategies, including the use of AFP as a drug carrier.

The authors of the manuscript describe in detail the structural features of recombinant AFP purified from HEK-293F cells, including the overall characteristics, glycosylation, metal binding, and ligands. Still, there is no possibility to verify the structural features since the whole manuscript lacks PDB and EMDB codes for obtained structures. A search in the PDB database showed that there is already deposited and published structure of human alpha-fetoprotein (PDB ID: 7YIM), also obtained by cryo-EM microscopy, with a resolution of 2.60Å.

The lack of access to structural data is particularly important when assessing the correctness of the metal and ligand binding sites. In the description of the experiment, the authors mention that zinc was not added to the sample and it is not included in the buffers and solutions used.

Respond: Very thank for the reminder of the reviewers. We have provided PDB and EMDB codes in the website. The AFP structural data has already been uploaded to the PDB website (<https://www.rcsb.org>), and the wwPDB validation report was viewed in attachment (PDB ID: 8X1N and EMDB ID: EMD-37997). Please check the AFP-PDB validation reports document.

Q1: Has the zinc content been confirmed in the test sample by additional tests, e.g. by absorption spectroscopy/fluorescence? Did the authors analyze the bond lengths and metal coordination in the obtained structure, e.g. via the CheckMyMetal server?

Respond: Many thanks for the reviewer's advise. According to reviewer's suggestion, we have supplemented the determination of the types and content of metal ions bound to AFP through experimental analysis. We detected the metal element content of both wild-type AFP and mutant AFP (H22A, H264A, H268A, and D280A) by inductively coupled plasma mass spectrometry (ICP-MS), and the result revealed the presence of numerous metal elements. Following data analysis, we presented a histogram illustrating the change in the molar ratio of metals to protein after AFP mutation (Supplementary Figure 4S). The findings indicated a reduction in the molar ratio of metal (magnesium, aluminum, nickel and zinc) to protein after mutation. This implies that the identified four amino acids in AFP possess the capacity to bind to different types of metal ions. Moreover, the molar ratio of some metal elements with AFP exceeds 1, it suggests the potential for AFP to bind to metal ions at multiple locations.

In addition, we analyzed the bond lengths and metal coordination via the CheckMyMetal server. The average distances of the zinc ion to three nitrogen atoms and one oxygen atom was 2.35 Å and 2.45 Å, respectively. Some parameters, such as Valence and nVECSUM, were not ideal. This could be due to the presence of multiple metal ions at this binding site, and the metal ion density being obtained through an averaging algorithm, leading to some deviation when a single metal element is fitted into this density.

In general, the manuscript is written in a clear way, with only a few editorial errors. The drawings are clear and aesthetic. Unfortunately, due to the inability to verify the correctness and quality of structural data, I cannot recommend this manuscript for publication.

Respond: Many thanks for the reviewer's guidance. We have provided PDB and EMDB codes in the website (<https://www.rcsb.org>). Please check the AFP-PDB validation reports document. We have supplemented and improved the revision of the manuscript according to the reviewer's requirements. Please check the new version of the manuscript.

Reviewer #2 (Remarks to the Author):

Alpha-fetoprotein (AFP) is a protein with crucial roles in fetal development and medical diagnostics. During pregnancy, it is produced by the developing liver and yolk sac of the fetus and can be detected in the mother's bloodstream. AFP serves as a significant diagnostic marker, with elevated levels indicating conditions such as liver cancer, germ cell tumors, and neural tube defects in fetuses. AFP is particularly important for the early detection and monitoring of hepatocellular carcinoma (HCC),

the most common type of liver cancer. It is suggested that AFP has immunosuppressive properties, which can inhibit immune cell activity. Thus AFP plays a vital role in fetal development and medical diagnostics, with growing potential as a target for cancer immunotherapies.

Authors present a cryoEM structure of human AFP determined at 3.6Å resolution. Protein expressed and purified from HEK293 F cells. They analyse the glycosylation by MS, and describe the FA and metal ion binding sites. They also make structural comparisons with highly similar HSA proteins.

A higher resolution structure of AFP has already been reported at 2.6Å (PDB ID 7YIM). Although it doesn't have any metal ions or glycans modeled.

Major comments:

There is no novelty in the structure as a higher resolution structure exists. Furthermore, multiple structures of highly similar HSA are also available and well described in the literature.

Respond: Very thank for the reviewer's guidance. The higher resolution structure of AFP (7YIM) was in our collaborative article (Nat Methods 2023;20: 123-130), but it merely involved the simple construction of a protein model, intended as a tool for validating the applicability of new materials, without conducting a detailed analysis of the AFP structure. In this manuscript, however, we conducted an in-depth analysis of AFP structural features, uncovering numerous characteristics of AFP that provide a structural basis for its physiological function and therapeutic strategies. These potential binding sites and modifications represent the novelty of this work.

HSA already has a considerable amount of structural data. Although it shares a high degree of sequence and structural similarity with AFP, in reality, there are still differences in their properties. In this manuscript, we compared various structural details of the two and pointed out the distinctions. These differences in amino acids may ultimately lead to changes in protein properties due to alterations in spatial conformation.

The authors emphasize the glycosylation analysis of heterologously expressed protein which might have no bearing or similarity with native AFP as the glycan composition could be totally different. This analysis would have made better sense on AFP purified from native sources.

Respond: Many thanks for the reviewer's reminder. It is indeed more relevant to analyze the glycosylation profile of AFP purified from native sources. Given that this study primarily focuses on the structural characteristics of AFP, it is more convenient and applicable to use the AFP protein expressed in vitro. The AFP obtained from the

HEK 293F eukaryotic expression system were the closest to the native human protein, and we finally elucidated the structure of recombinant AFP. Our objective in analyzing the glycosylation profile serves a dual purpose: it serves as biochemical validation of AFP glycosylation structures and facilitates comparative analysis of the glycosylation disparities in AFP expressed by different cells in vitro. The heterogeneity in glycan composition arises primarily from the stochastic nature of its addition, implicating that the glycosylation profile is largely influenced by the cellular milieu interior during synthesis. Consequently, our analysis focused on the glycosylation modifications of AFP expressed in both hepatocellular carcinoma cells (Bel7402) and non-hepatocellular carcinoma cells (HEK 293F). The aim was to observe the glycosylation profile of N-linked glycans in AFP synthesized by different cell types, thereby laying the groundwork for further investigations into the glycosylation structure and its function in AFP within hepatocellular carcinoma patients in vivo.

The authors discuss differences in conformation with HSA but again physiological relevance of these differences is not clear. Moreover, the differences could be due to variations in protein preparation/buffer conditions.

Respond: Thanks the reviewer's guidance. The physiological differences between AFP and HSA have been previously reported, and the structural characteristics we analyzed can partially account for these physiological distinctions. For instance, the disparity in the number and affinity of ligands binding to HSA and AFP (PMID: 91613, PMID: 2423073, PMID: 2472294, and PMID: 2441917) is due to the differences in the amino acid composition of their binding pockets, causing AFP and HSA to exert their respective physiological functions in different temporal and spatial contexts within the human body, such as predominance of AFP in the blood of newborns and liver cancer patients, while HSA predominates in the blood of healthy adults. Furthermore, when selecting HSA structures for comparison with AFP, we particularly focused on the preparation conditions of HSA. Firstly, the fatty acid bound to HSA is palmitic acid same as AFP. Secondly, when comparing metal ions, we specifically chose the HSA-Zn²⁺ structure. Lastly, both HSA and AFP were used in PBS buffer solutions. Please check the new version of the manuscript.

Druggability of the FA binding sites is discussed, however, no evidence is provided for the same.

Respond: In our discussion, we mentioned the druggability of AFP, mainly based on the clinical application of the HSA-drug. Due to the similar binding pockets in the structures of AFP and HSA, we speculate that exploring the druggability of AFP is highly feasible. Nevertheless, we will continue to invest time in verifying the druggability of AFP in the subsequent stage.

Similarities and differences between the metal ion and FA binding sites between HSA and AFP are discussed. Nevertheless, it's important to note that these findings remain speculative as no mutational analysis has been conducted to confirm their validity.

Respond: Thanks the reviewer's guidance. Comparing the similarities and differences between the two protein structures is achieved through a comparison based on the existing conformations of the two proteins. Of course, due to the fact that the sample preparation processes are not entirely identical, there inevitably exists a certain degree of speculation. To verify the metal ion binding site, we conducted additional tests. We used inductively coupled plasma-mass spectrometry(ICP-MS) to detect the metal element content of wild-type AFP and mutant AFP (H22A, H264A, H268A, and D280A). The results indicate that AFP does indeed bind to metal ions (Supplementary Figure 4S B). We analyzed the metal coordination via the CheckMyMetal server, and found the metal ions were tetrahedral coordinated with the side chains of these four amino acids. Therefore, this position is indeed AFP-metal ion binding site. Finally, the metal ion binding sites between AFP and HSA are clearly different.

The FA bound by AFP has been identified as palmitic acid using gas chromatography mass spectrometry, hence, we chose to compare it with the structure of HSA bound with palmitic acid. Due to the numerous amino acids associated with the FA binding sites, it is not suitable to use amino acid mutation methods for binding validation. Additionally, a suitable method for identifying the specific FA bound in each pocket has not yet been found. In the future, we will explore alternative methods for validation. The comparison of the FA binding sites between AFP and HSA primarily provides some structural foundations for the exploration of AFP drug properties in the future.

Minor:

RMSD values of HSA and AFP protein should be provided.

Respond: Very thank the reviewer's guidance. The RMSD of HSA and AFP protein is 1.186Å between 249 pruned atom pairs (across all 571 pairs: 5.763Å). Please check the new version of the manuscript.

Page 13 line 380 words effective and effect are used in the same sentence. May be modified or replaced with a synonym.

Respond: We have revised the manuscript according to the reviewer's requirements. Please check the new version of the manuscript.

The authors have frequently emphasized the elucidation of the structure of AFP, specifically using the terms "actual structure" or "true structure." It is worth noting that they have determined the structure of a recombinantly expressed protein through the application of single-particle cryo-EM, an averaging method. Therefore, the terms "actual" and "true" should be removed.

Respond: We have revised the manuscript according to the reviewer's requirements.
Please check the new version of the manuscript.

REVIEWERS' COMMENTS:

Reviewer #2 (Remarks to the Author):

The authors have addressed my concerns. However, several grammatical and typographical errors remain. Authors should carefully check the entire manuscript and remove them.

The authors have responded adequately to the concerns of R1 as well. The ICP-MS experiments on wild type and histidine mutants does suggests binding of Zn alongwith other metals.

Respond to the Reviewers' comments

REVIEWERS' COMMENTS:

Reviewer #2 (Remarks to the Author):

The authors have addressed my concerns. However, several grammatical and typographical errors remain. Authors should carefully check the entire manuscript and remove them.

Respond: Many thanks to the reviewers for their positive comments on our manuscript. We have revised the grammatical and typographical errors of the manuscript as required by the reviewers. Please check the new revised version of the manuscript.

The authors have responded adequately to the concerns of R1 as well. The ICP-MS experiments on wild type and histidine mutants does suggests binding of Zn alongwith other metals.

Respond: Very thank for the reviewers positive comments.